# Flexible and durable wood-based triboelectric nanogenerators for self-powered sensing in athletic big data analytics

Jianjun Luo [1,2,5], Ziming Wang[1,2,5], Liang Xu [1,2,5], Aurelia Chi Wang[3], Kai Han[1,2], Tao Jiang[1,2], Qingsong Lai[1,2], Yu Bai[1,2], Wei Tang[1,2], Feng Ru Fan[4]* & Zhong Lin Wang [1,2,3]*

In the new era of internet of things, big data collection and analysis based on widely distributed intelligent sensing technology is particularly important. Here, we report a flexible and durable wood-based triboelectric nanogenerator for self-powered sensing in athletic big data analytics. Based on a simple and effective strategy, natural wood can be converted into a high-performance triboelectric material with excellent mechanical properties, such as 7.5-fold enhancement in strength, superior flexibility, wear resistance and processability. The electrical output performance is also enhanced by more than 70% compared with natural wood. A self-powered falling point distribution statistical system and an edge ball judgement system are further developed to provide training guidance and real-time competition assistance for both athletes and referees. This work can not only expand the application area of the self-powered system to smart sport monitoring and assisting, but also promote the development of big data analytics in intelligent sports industry.

[1] CAS Center for Excellence in Nanoscience, Beijing Key Laboratory of Micro-nano Energy and Sensor, Beijing Institute of Nanoenergy and Nanosystems, Chinese Academy of Sciences, Beijing 100083, P. R. China. [2] School of Nanoscience and Technology, University of Chinese Academy of Sciences, Beijing 100049, P. R. China. [3] School of Material Science and Engineering, Georgia Institute of Technology, Atlanta, Georgia 30332, USA. [4] Flex Laboratory, School of Industrial Engineering, Purdue University, West Lafayette, Indiana 47907, USA. [5] These authors contributed equally: Jianjun Luo, Ziming Wang, Liang Xu *email: fan296@purdue.edu; zlwang@gatech.edu

With a rapid development of internet of things (IoTs)[1,2] and big data[3,4] over the last few decades, a wide range of application domains, such as home automation, healthcare, security, environmental monitoring, and information communication, have undergone revolutionary changes[5–9]. The sports field has also been greatly influenced by the technical advances, entering the digital age. Big data services, including exercise performance, health data, training statistics, and analysis, can effectively help athletes in daily training and developing game strategies, and are becoming an indispensable means for winning competitions[10]. Real-time data acquisition relies on widely distributed sensors, which are generally powered by conventional energy storage devices, such as batteries. Considering their limited lifetime, high replacement or recharging costs, and environmental issues, it is highly desirable to develop a sustainable and maintenance-free sensing technology.

Recently, triboelectric nanogenerators (TENGs), based on the coupling effect of contact electrification and electrostatic induction, have been developed into a powerful technology for converting mechanical energy into electricity, with unique advantages of low cost, high efficiency, simple structure, and diverse material options[11–17]. To provide sustainable power source for electronics, a concept of self-charging power system has been developed by integrating TENGs with energy storage devices into one single unit[18–22]. In addition, by directly converting mechanical stimuli to electrical signals, TENGs can also operate as self-powered sensors for pressure, tactile, or motion sensing without extra power supply, which is vital for developing maintenance-free systems[23–27]. Therefore, the TENG technology can be an effective power solution for the new era—the era of IoTs, sensor networks, robotics, and artificial intelligence, where large amounts of distributed devices should be applied.

Currently, most of the materials used in TENGs are non-degradable synthetic polymers, which could possibly result in severe environmental pollution[28]. Natural wood is one of the most abundant resources on earth. As a renewable, sustainable, and biodegradable material, it has been widely used in buildings, energy storage, water purification, and flexible electronics[29–35]. However, the mechanical and triboelectric properties of natural wood are unsatisfactory for fabricating TENGs. Therefore, an effective processing method that can adjust the properties of the natural wood for fabricating a high-performance TENG will provide a great opportunity for developing sustainable and eco-friendly self-powered system.

Herein, we report a simple and effective two-step process for fabricating wood films with excellent mechanical and triboelectric properties, enabling flexible, durable, and high-performance wood-based triboelectric nanogenerators (W-TENGs). The flexible and durable W-TENG is capable of outputting a transferred charge density of 36 μC m$^{-2}$, which is more than 70% higher than that of the TENG based on natural wood. Meanwhile, this W-TENG possesses other merits such as light weight (0.19 g), thin thickness (0.15 mm), and cost effective. Owing to its excellent performance, the W-TENG is further utilized as an active wood-based triboelectric sensor (W-TES) for fabricating a smart ping-pong table. A self-powered falling point distribution statistical system that can perform velocity sensing, motion path tracking, and distribution statistics to assist athletes in training, as well as a self-powered edge ball judgement system that can help the referee's decision are demonstrated. This work opens up new avenues for self-powered systems in intelligent athletic facilities and big data analytics, and may profoundly influence the global sports field.

## Results

**Smart sports equipment enabled by the W-TENG.** Figure 1a shows a schematic of the fabrication process of our flexible and durable wood-based TENG. The natural wood was firstly converted into flexible wood using a top-down two-step approach involving partially removal of lignin/hemicellulose from natural wood followed by hot-pressing. Then, the obtained flexible wood was used for fabricating the TENG device (see Methods). Considering the wide use of wood materials in sports industry, the W-TENG will have great potential for building self-powered sensing system in sports equipment. Taking table tennis as an example, the single-electrode mode W-TENG can be applied to fabricate a smart ping-pong table, for a table tennis training and assisting system, as schematically shown in Fig. 1b. This system possesses two capabilities: the first is falling point distribution statistics for athletic big data analytics, and a W-TENG array will be installed on the surface of the table in order to accurately identify the impact point of the ping-pong ball; the second function is disputed edge ball judgement, with two single-electrode mode W-TENGs setting on the top edge and side edge of the table respectively. The wood material in W-TENG can keep the basic properties of the table surface, which is significant for sports apparatus. Figure 1c shows the scanning electron microscopy (SEM) image of the flexible wood. Photograph of two flexible and durable single-electrode mode W-TENGs is shown in Fig. 1d, demonstrating its excellent flexibility. Besides, our W-TENG owns other merits such as light weight and thin thickness, which can be only 190 mg (area, 4 × 4 cm$^2$) and 150 μm (Fig. 1e, Supplementary Fig. 1).

**Wood treatment and characterization.** Chemical treatment plays a key role in improving the mechanical and triboelectric properties of the natural wood. Owing to different stabilities of three wood components in NaOH/Na$_2$SO$_3$ solution, substantial lignin/hemicellulose content can be removed, but only modest cellulose content will be dissolved after chemical treatment. As graphically illustrated in Fig. 2a, the chemical treatment leads to structural changes to the natural balsa wood. The cross-sectional morphology and microstructure of the wood samples were characterized by scanning electron microscopy (SEM). After chemical treatment, the 3D porous wood structure evolves from open latticed cell lumina to crumpled ones with shrunken diameters and irregular shape (Fig. 2b, c). Besides, the wood cell walls become porous and less rigid due to partial removal of lignin/hemicellulose, as shown in the enlarged cross-sectional SEM images in Supplementary Fig. 2. Such chemical treatment is beneficial and essential to the following hot-pressing process for further TENGs fabrication.

To validate the component change, Fourier transform infrared spectroscopy (FTIR) of the wood samples before and after chemical treatment was carried out. The FTIR spectra of the natural wood and treated wood show that the groups assigned to lignin/hemicellulose were substantially reduced by the mixed solution of NaOH/Na$_2$SO$_3$ (Fig. 2d). Meanwhile, the weight changed from 100% to 58% (Supplementary Fig. 3), also indicating the removal of the lignin/hemicellulose component.

Figure 2e compares the tensile stress-strain curves for the natural wood and treated wood. Both curves show a linear deformation behavior before tensile failure. The treated wood demonstrates a high tensile strength of 284.2 MPa, which is 7.5 times of that for the untreated natural wood. Furthermore, the tensile strength of the treated wood only decreases slightly after bending 1000 cycles, suggesting its excellent flexibility and durability (Fig. 2e, Supplementary Fig. 4). As demonstrated in Fig. 2f, the natural wood film breaks easily upon bending, while

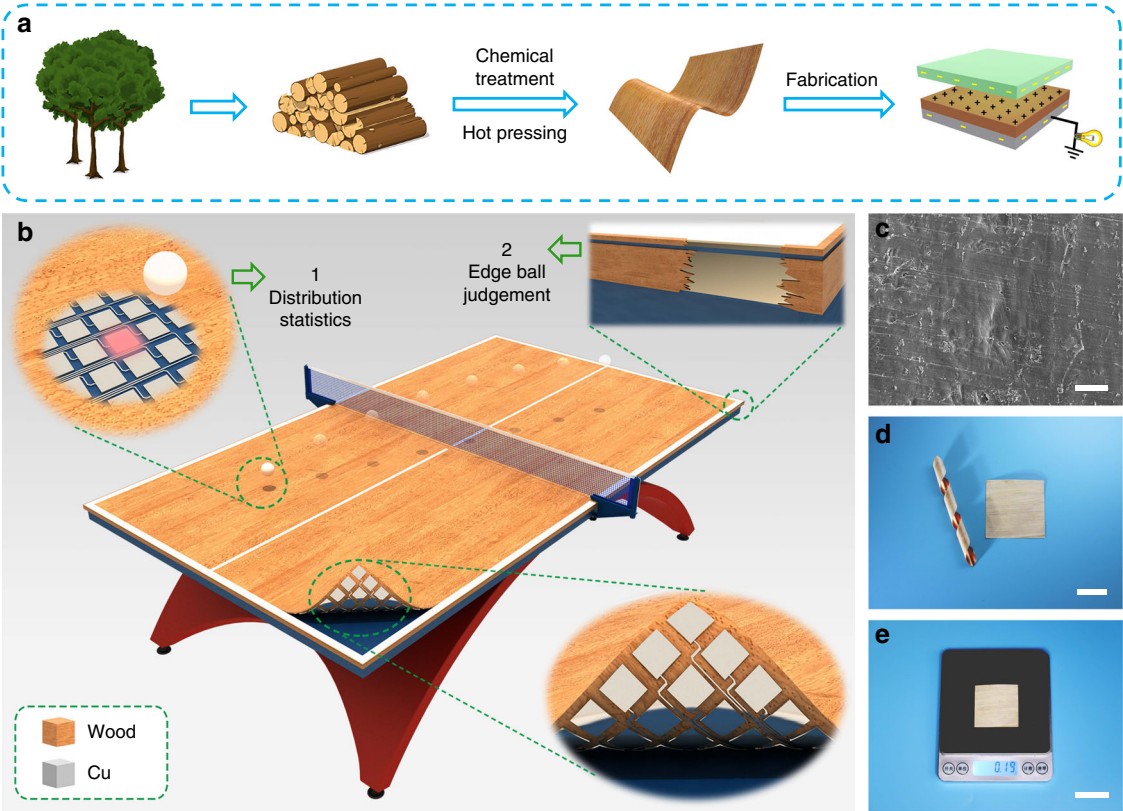

**Fig. 1** Fabrication and schematic of the flexible wood-based TENG and smart ping-pong table. **a** Diagram of the process for fabricating a flexible W-TENG. **b** Schematic illustration of the natural wood-based smart ping-pong table. **c** Scanning electron microscopy (SEM) image of the treated wood surface. Scale bar, 20 μm. **d** Photograph of the as-prepared W-TENG devices, demonstrating its excellent flexibility. Scale bar, 2 cm. **e** Weight of the W-TENG device. Scale bar, 3 cm

the mechanical flexibility of the wood film can be enormously improved after treatment.

Meanwhile, the wear resistance of the treated wood was also studied through measuring the coefficient of friction (COF), as shown in Fig. 2g. During 1000 cycles of rubbing against a steel ball, the COF of the natural wood increased with increasing cycles. However, the COF of the treated wood remained stable, suggesting very little abrasion after rubbing test. This can be further confirmed by directly observing the surface topography of the two kinds of wood after rubbing test (Fig. 2h). Obvious wear trace can be seen on the natural wood after the test, while almost no notable wear features on the treated wood can be observed, demonstrating its great improvement in anti-wear capability after chemical treatment and hot-pressing.

Interestingly, the processability of the wood film can also be greatly improved after treatment. As shown in Fig. 2i, the natural wood slice needs a large pressure of ~180 MPa to be compressed from 1.5 to 0.12 mm. By contrast, the wood slice with chemical treatment can be easily compressed into the same thickness under a much smaller pressure of ~10 MPa. Moreover, its surface area can be further increased by combining template method with hot-pressing, which is important for improving the output performance of the TENG[36,37]. Figure 2j and Supplementary Fig. 5 show the surface interferometer images of the treated wood after hot-pressing using various sand paper as the mold, indicating significant increase in the surface area with micro structures. Furthermore, the treated wood is shown to be more resistant to humidity from the accelerated tests (Supplementary Fig. 6).

**Performance of the W-TENG.** Based on the treated wood as dielectrics, TENGs with different working modes can be

fabricated. If the back Cu electrode is connected to the ground by a metal wire through an external load, the W-TENG will work in the single-electrode mode, which is especially suitable for sensing the moving object in sports field. Taking PTFE as the moving object, the working mechanism of W-TENG in single-electrode mode is illustrated in Fig. 3a. Once the PTFE film contacts with the treated wood film, negative triboelectric charges are gained by the PTFE due to its stronger ability to capture negative charges, whereas the wood is left positive charged (i). Once the PTFE begins to separate with the wood film, the potential difference between two surfaces will gradually increase, leading to an instantaneous electron flow from the ground to the Cu electrode in the external circuit (ii). This transient flow of the electrons continues until the PTFE and wood film are totally separated (iii). When the PTFE film is approaching back to the wood film, the electrons will be repelled back from the Cu electrode to the ground through the external load (iv). By repeating the contact-separation movement between the moving dielectric object and the W-TENG, an alternative current will be generated. Corresponding simulations of potential distribution in three different states by COMSOL are presented in Fig. 3b.

To characterize the output performance of the W-TENG fabricated with treated wood, a commercial PTFE film was used as the moving object to have contact-separation movement relative to a W-TENG device (area, $3 \times 3$ cm$^2$). Unless otherwise specified, the acceleration (7.5 m s$^{-2}$), frequency (1 Hz), and pressing force (20 N) of the contact-separation motion are controlled to be the same by a linear motor for all following tests. The open-circuit voltage ($V_{OC}$), short-circuit current ($I_{SC}$), and transferred charge density ($\Delta\sigma$) of the W-TENG based on treated wood are shown in Fig. 3c, d, and Supplementary Fig. 7,

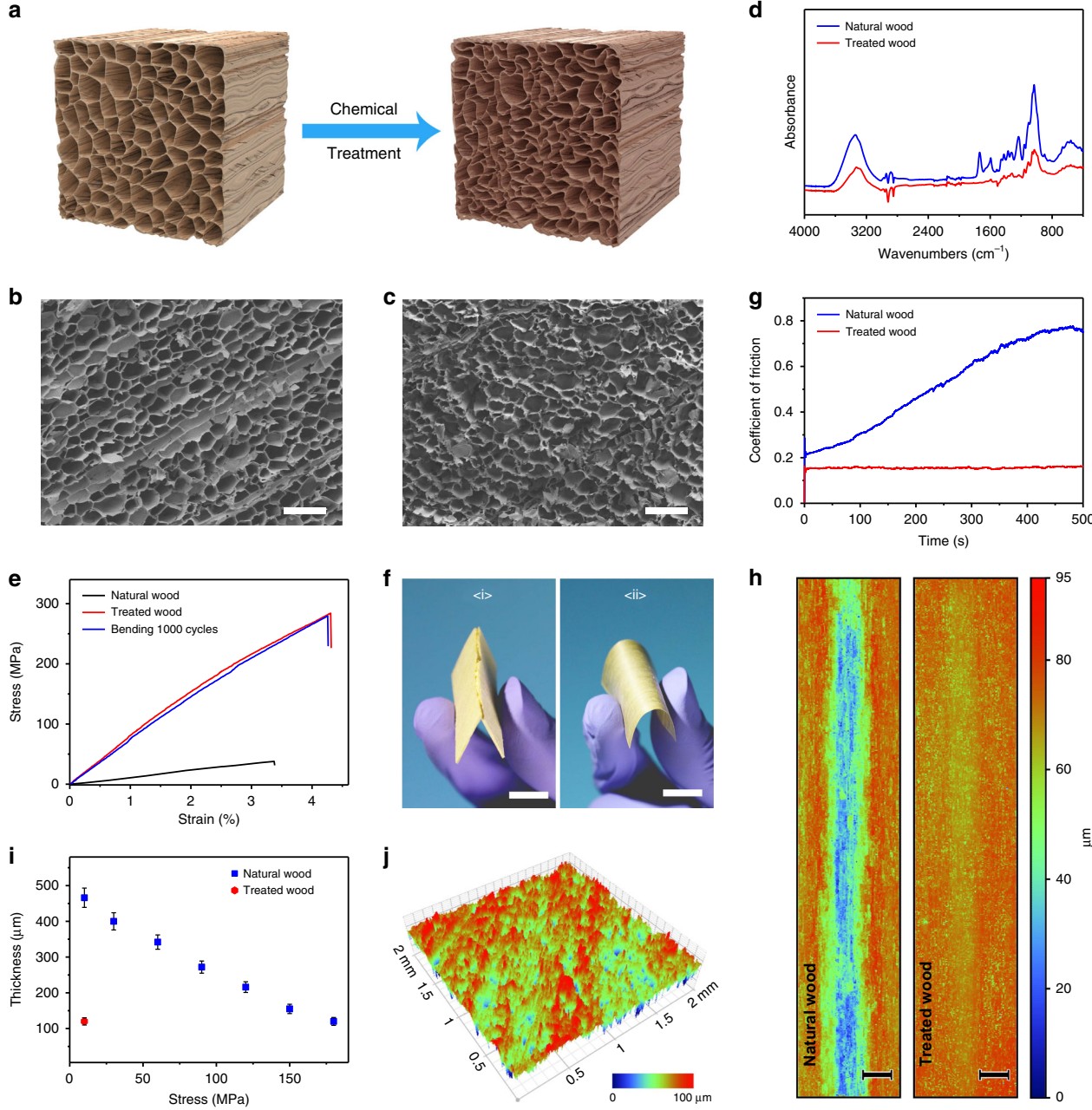

**Fig. 2** Evolution of natural balsa wood upon chemical treatment and hot-pressing. **a** Graphical illustration of the structural evolution of natural wood upon chemical treatment. **b**, **c** Cross-sectional SEM images of the natural wood and treated wood. Scale bar, 100 μm. **d** FTIR spectra of the natural wood and treated wood. **e** Tensile stress-strain curves of the natural wood and treated wood (before and after bending 1000 cycles). **f** Comparison of flexibility between natural wood and treated wood. (i) Natural wood film breaks upon bending. (ii) Treated wood shows high flexibility upon bending. Scale bar, 1 cm. **g** Coefficient of friction evolution of natural wood and treated wood. **h** Interferometer images of wear traces on natural wood and treated wood after 5000 cycles of rubbing, showing notable decrease of wear depth of the treated wood. Scale bar, 500 μm. **i** Compression thickness of the natural wood and treated wood after hot-pressing. Error bars indicate standard deviations for 5 sets of data points. **j** Interferometer image showing the surface topography of treated wood after hot-pressing using 600-grit sand paper as the mold

with peaks of 81 V, 1.8 μA, and 36 μC m$^{-2}$, respectively. To investigate the enhancement of the triboelectric property of the wood after treatment, output performance of the W-TENG based on contact-separation mode are compared (Details are discussed in Supplementary Fig. 8, Supplementary Note 1). When using treated wood for fabricating the W-TENG, it can be found that the $V_{OC}$, $I_{SC}$, and $\Delta\sigma$ are all improved greatly (Fig. 3e). By comparing their transferred charge density, which is the most effective parameter for evaluating output performance, the TENG based on treated wood is 71% higher than that of the natural

wood. It is worth noting that the wood films used in above comparison tests are of the same thickness. Therefore, this output performance enhancement can be attributed to the surface chemical composition and structure changes in the wood samples. Owing to the substantial removal of lignin/hemicellulose, cellulose will be exposed on the wood surface, which is more positive in triboelectric series than other wood content, leading to an increase in surface charge density during friction process[38,39]. When using different kinds of wood, similar triboelectric property enhancement could be observed, proving the

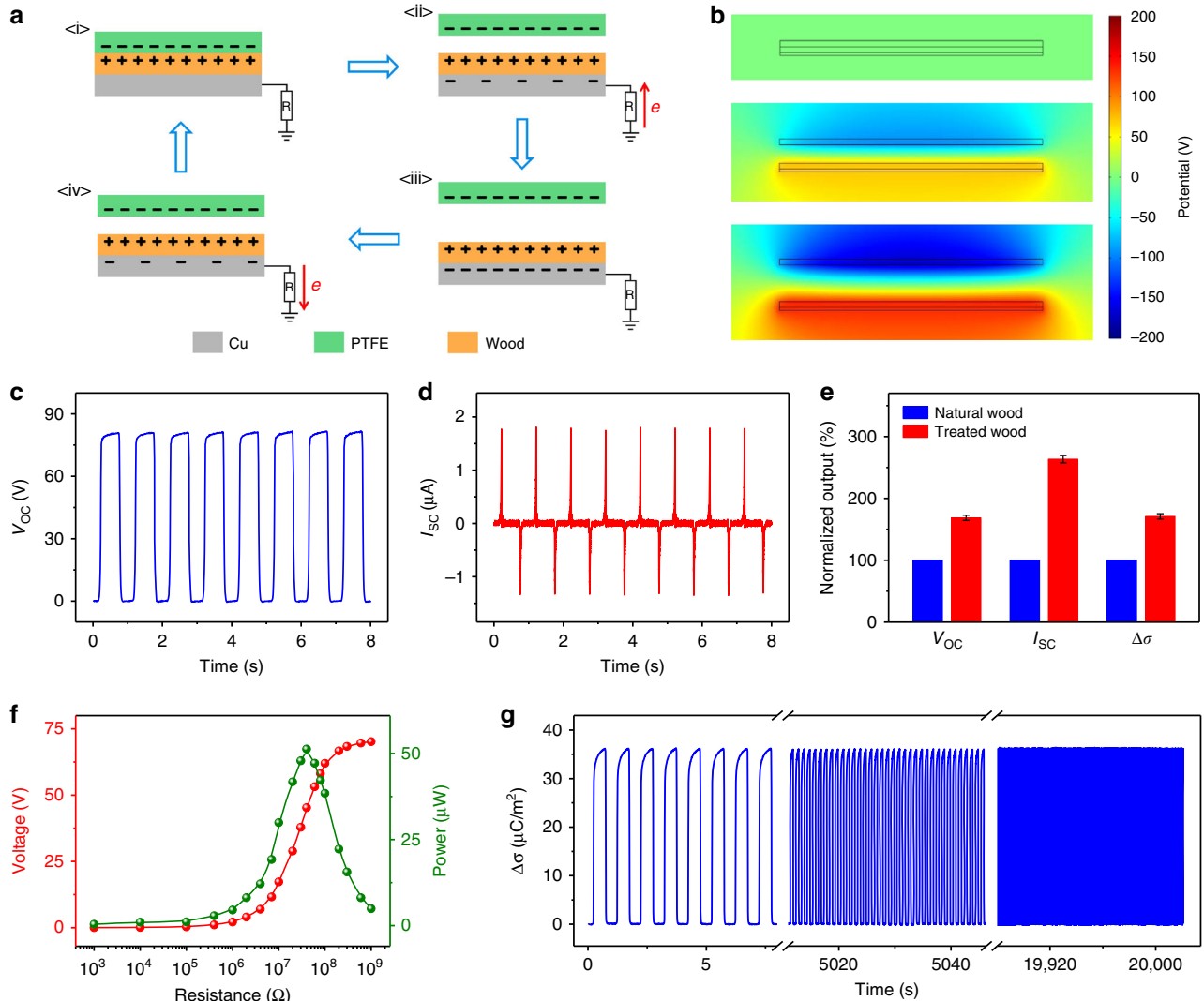

**Fig. 3** Working mechanism and output performance of W-TENG in single-electrode mode. **a** Schematics of the operating principle for the W-TENG. **b** Potential simulation by COMSOL to elucidate the working principle. **c**, **d** Open-circuit voltage, short-circuit current of the W-TENG based on treated wood. **e** Comparison of the output performance between W-TENGs based on natural wood and treated wood. **f** Dependence of the output voltage and peak power of the W-TENG based on treated wood on the resistance of external load. **g** Stability and robustness measurement of the W-TENG, where the transferred charge density was recorded for over 20,000 cycles at a frequency of 1 Hz

universality of our wood treatment method (Supplementary Fig. 9). With the increase of acceleration, the $I_{SC}$ of the W-TENG shows an almost linear rise (Supplementary Fig. 10).

To evaluate the effective output performance of the W-TENG based on treated wood, the output voltage was measured with various resistances applied as the external load. The relationship between the output voltage/power and the resistance is plotted in Fig. 3f. Under an external load resistance of 40 MΩ, the maximum peak output power of 51 μW (57 mW m$^{-2}$ in areal power density) can be achieved. Supplementary Fig. 11 shows the charging curves of the capacitor when charged by the W-TENG devices, also indicating that the wood treating method can greatly improve the output performance of the W-TENG. As shown in Fig. 3g, the transferred charge density only has a little decay in continuous operation of 20,000 cycles, confirming superior stability of our W-TENG. The output performance of the W-TENG in different humidity is also investigated (Supplementary Fig. 12). When contacting with different materials, the output of the W-TENG is summarized in Supplementary Fig. 13. As demonstrated above, the excellent properties of the treated wood

as dielectrics facilitate the integration of TENG with various sports equipment that usually made of wood material, while high performance of TENGs can still be obtained. This can render smart features to the athletic apparatus that provide rich information in the process of sports.

**Self-powered W-TES for athletic big data analytics.** With the development of modern table tennis, the sense and ability for controlling the falling point of the ball are becoming increasingly important as the outcome of the match would be heavily influenced by them. To improve the athletic training efficiency and quality, a self-powered falling point distribution statistical system was built based on the W-TENG, where the ping-pong ball itself can be regarded as the moving object for it can get effectively triboelectric charged during the competition. In such case, the W-TENG operates as a self-powered W-TES. Owing to the hardness enhancement of the treated wood, the W-TENG has almost no influence on the elastic property of the table surface (Supplementary Fig. 14). Supplementary Fig. 15 shows the sensing mechanism of this W-TES in five different states during a ping-

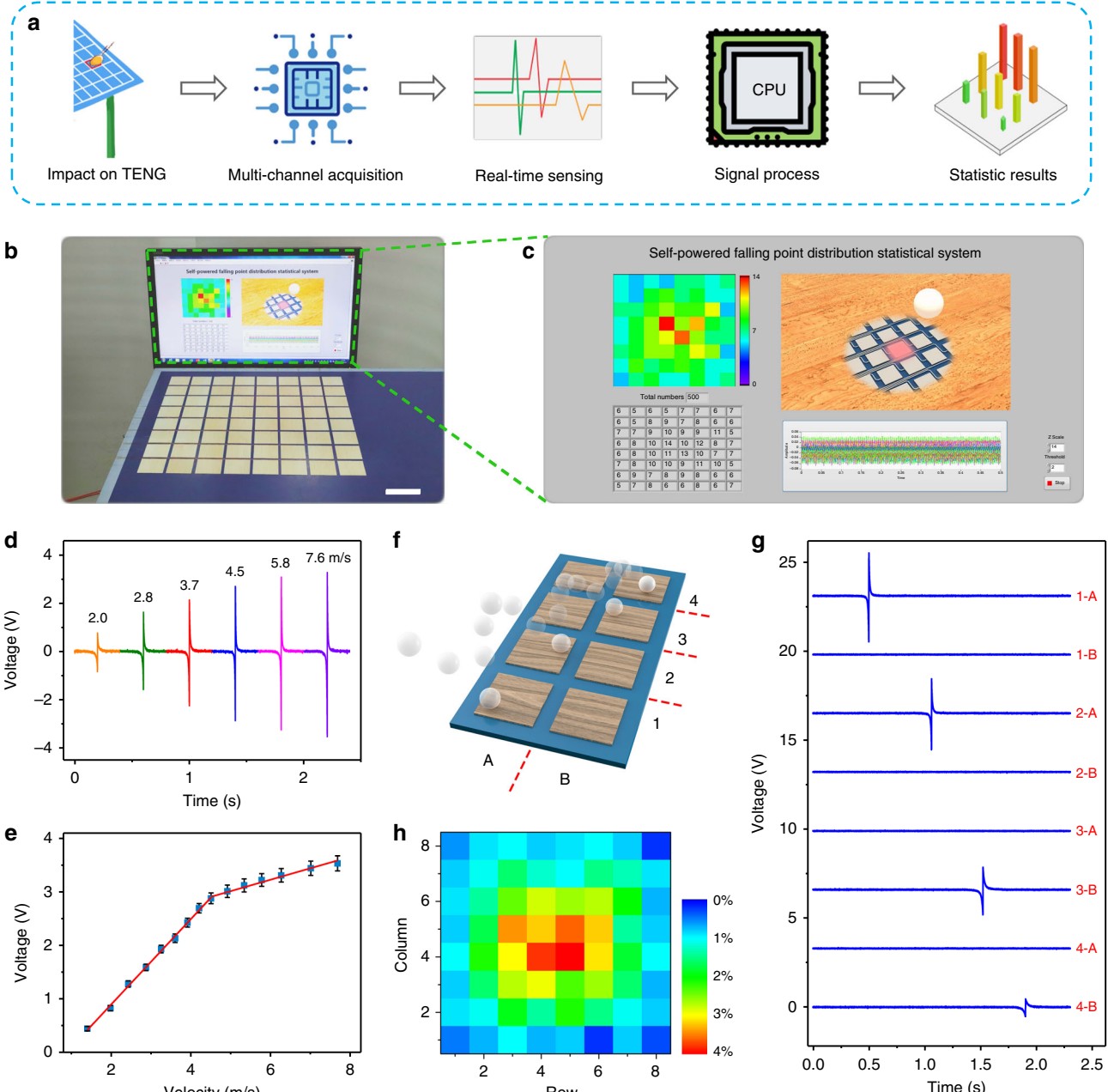

**Fig. 4** Application of the W-TENG in a self-powered falling point distribution statistical system. **a** Scheme diagram of the W-TENG based self-powered falling point distribution statistical system. **b** Demonstration of the self-powered falling point distribution statistical system. Scale bar, 6 cm. **c** Screenshot showing the real-time statistical result of the self-powered system. **d** The measured output voltage of the TENG under the impact of ping-pong balls with variable velocities. **e** The summarized relationship and linear fitting between output voltages and velocities. Error bars indicate standard deviations for five sets of data points. **f** Graphical illustration showing the moving path of the ping-pong ball above a 4 × 2 W-TES array. **g** Real-time output voltage signals when a ping-pong ball impacted the surface of a 4 × 2 W-TES array along a path: 1-A → 2-A → 3-B → 4-B. **h** 2D mapping figure showing the distribution of statistical percentage of the falling points

pong ball impacting on it. As a proof-of-concept demonstration, a 8 × 8 W-TES array was fabricated and set on the surface of the table for precisely detecting the falling point (Supplementary Fig. 16). The size of each W-TES unit is 4.5 × 4.5 cm², and the gap between two adjacent units is 0.5 cm. Here, a synchronous data acquisition card (PXIe-4300, National Instruments) with integrated signal conditioning was used for multi-channel output voltage measurements. The equivalent circuit of this measuring system is illustrated in Supplementary Fig. 17. As depicted in Fig. 4a, while a ball impacts on the table surface, the W-TES at the impact position will generate an obvious output signal. Through

the multi-channel data acquisition method, real-time voltage signals of each sensor unit could be detected at the same time. After signal processing, real-time statistic results can be displayed in the program. Figure 4b shows the self-powered falling point distribution statistical system. The enlarged view of this system is shown in Fig. 4c, including a mapping figure of distribution, statistic results of each pixel, and real-time voltage signals of all the channels. For demonstration, a ping-pong ball was used to impact on different positions of the sensor array for five times. Each impact can cause a corresponding update in the mapping figure and generate statistic results in each pixel, illustrating

excellent performance of this system in sensing falling point location and statistical distribution. (Supplementary Movie 1).

The W-TES can also work for ball velocity sensing without an external power source. Fig. 4d is the real-time measurement result of output voltage under different ball approaching velocities. As the velocity of the ping-pong ball increases, the output voltage shows a clear increasing trend. The relationship between output voltage signal and the magnitude of velocity is plotted in Fig. 4e. It is worth noting that the curve can be divided into two distinct regions. In the low-velocity region ($<4.5\,\mathrm{m\,s^{-1}}$), a high velocity sensitivity of $0.78\,\mathrm{V/(m\,s^{-1})}$ is achieved with excellent linearity ($R^2 = 0.997$). In the region beyond $4.5\,\mathrm{m\,s^{-1}}$, the velocity sensitivity drops to $0.21\,\mathrm{V/(m\,s^{-1})}$, but still has good linearity ($R^2 = 0.967$). The excellent linear relationship indicates that the W-TENG could be used as a sensitive self-powered sensor for measuring the magnitude of ball velocity. Besides, the W-TES exhibits a rapid response time of $<25\,\mathrm{ms}$ (Supplementary Fig. 18). Moreover, our W-TES array can also be used for ball motion path tracing. Fig. 4f illustrates the moving path of a ping-pong ball on a $4 \times 2$ pixelated W-TES array (1-A → 2-A → 3-B → 4-B). The corresponding output voltage signals with time of each channel are shown in Fig. 4g, indicating the feasibility of our sensing system for motion path tracing.

Supplementary Movie 2 shows a long period falling point distribution test using the self-powered falling point distribution statistical system, demonstrating the stability and accuracy of our system. In order to further verify its ability of big data analysis for sports, 10,000 times impacts of ping-pong ball were fully collected and statistical probability for each pixel was recorded in Fig. 4h and Supplementary Fig. 19. By analyzing the statistical result, the exercise habit data of athletes can be obtained for further guiding their training and developing better competition strategies. This prototype of W-TENG based self-powered falling point distribution statistical system shows potential for creating a new type of sports equipment for athletic training.

**Self-powered W-TES as an edge ball judgement system.** Besides the construction of falling point distribution statistical systems for assisting athletes' trainings, improving the accuracy of judgement in table tennis competition also plays a crucial role for helping referees' decisions. Edge ball often happens during the table tennis match, whose judgement result could easily arouse controversy. It is especially difficult to distinguish whether it is top edge ball or side edge ball. Considering the excellent mechanical property and high output performance of the W-TENG, we applied it in constructing a self-powered edge ball judgement system. Figure 5a illustrates the operating mechanism of our self-powered edge ball judgement system. When edge ball happens, the mechanical agitation can be transformed into electrical signal and synchronously recorded by the W-TENGs attached on the top and side edge of the table, respectively. After signal filtering and analog-to-digital (A/D) conversion, the real-time judgement result can be displayed to evaluate whether the edge ball is effective. Figure 5b shows the photograph of the two W-TENGs attached on the edge of the table. Benefit from its excellent flexibility, the W-TENG can easily fit on the surface of the edge.

To demonstrate the feasibility of the W-TENG based self-powered edge ball judgement system, we test two different edge balls using our self-powered judgement system. Figure 5c, f are photographs showing top edge ball and side edge ball, respectively. Corresponding trajectory of the ball in two different circumstances can be seen in Supplementary Movie 3 and 4. When the ball impacts on the top edge of the table, an obvious output signal will be generated by TENG-1 with subsequent data processing (Fig. 5d). By numeric comparison of the synchronous

voltage output of the two TENGs, real-time judgement result will be displayed in the program interface (Fig. 5e). Similarly, while side edge ball occurs, TENG-2 will generate a distinct output signal (Fig. 5g), and then related judgement result will be displayed (Fig. 5h). With such judgement system, two kinds of edge balls can be determined accurately in real time (Supplementary Movie 5). This demonstration presents potential applications in providing self-powered real-time referee with low cost and high accuracy for athletic contest.

## Discussion

In conclusion, we developed a flexible and durable high-performance wood-based triboelectric nanogenerator for self-powered sensing in sports equipment and athletic big data analytics. The high-performance wood material was prepared using a two-step method involving a boiling process in aqueous mixture of NaOH and $Na_2SO_3$ followed by hot-pressing. The triboelectric and mechanical performance of the wood, including tensile strength, flexibility, wear resistance, and processability, can be greatly improved after the treatment. More importantly, a significant output performance enhancement of ~70% can be obtained, when using the treated wood for fabricating W-TENG. As demonstrated, the W-TENG is adopted to fabricate a smart ping-pong table with multiple sensing functions. To provide effective training evaluation and guidance for athletes, a self-powered falling point distribution statistical system was realized, and real-time training data can be recorded for big data analytics. Furthermore, a self-powered edge ball judgement system was also constructed for assisting referees' decisions in real time. This research demonstrates applications of environmental-friendly W-TENGs, which are expected to bring a great opportunity in athletic big data analytics and open up a new field of wood-based electronics combining with the self-powered system.

## Methods

**Materials and chemicals.** Balsa wood was used for the fabrication of flexible wood. Sodium hydroxide (>97%, Sigma-Aldrich), sodium sulfite (>98%, Sigma-Aldrich), and deionized (DI) water were used for processing the wood.

**Two-step process towards flexible wood.** First, natural balsa wood slices (typical sample dimension: $30.0\,\mathrm{mm} \times 30.0\,\mathrm{mm} \times 1.5\,\mathrm{mm}$) were immersed in a boiling mixed aqueous solution of 2.5 M NaOH and 0.4 M $NaSO_3$ for 7 h, followed by immersion in boiling DI water for several times to remove the chemicals. Next, the wood slices were pressed at 100 °C under a pressure of ~10 MPa (typically) for ~2 h to obtain the flexible wood (typically $30.0\,\mathrm{mm} \times 30.0\,\mathrm{mm} \times 0.1\,\mathrm{mm}$).

**Fabrication of the flexible and durable wood based TENG.** The obtained flexible wood was cut into desired shapes as the dielectric electrification layer. A layer of Cu film with the same size was pasted on the flexible wood as the electrode to construct the single-electrode mode TENG. For the contact-separation mode TENG, a PTFE film was adhered to the second Cu film as another triboelectric material and electrode respectively. Cu wires were attached to each Cu film for electric connection. In the fabricating of natural wood based TENG, the flexible wood is replaced with natural wood.

**Characterization and measurements.** The W-TENG was driven by a linear motor (Linmot E1100) for electrical measurements. A programmable electrometer (Keithley 6514) was used to test the open-circuit voltage, short-circuit current, and transferred charges. For multichannel voltage measurements of the triboelectric sensor array in the smart ping-pong table, a synchronous data acquisition card (PXIe-4300, National Instruments) with integrated signal conditioning was used. The software platform was constructed on the basis of LabVIEW, which is capable of realizing real-time multi-channel data acquisition and analysis. A scanning electron microscope (SU8020, Hitachi) and an optical profilometer (Contour GT-K, Bruker) was used to characterize the morphologies of the wood samples. A Fourier transform infrared spectroscope spectrometer (VERTEX80v, Bruker) was used to measure the FTIR spectrum. Mechanical tensile properties of the wood samples were measured using a ESM301/Mark-10 tester. The samples were clamped at both ends and stretched along the sample length direction until they fractured with a constant strain rate of $4\,\mathrm{mm\,min^{-1}}$. Wear experiments of the

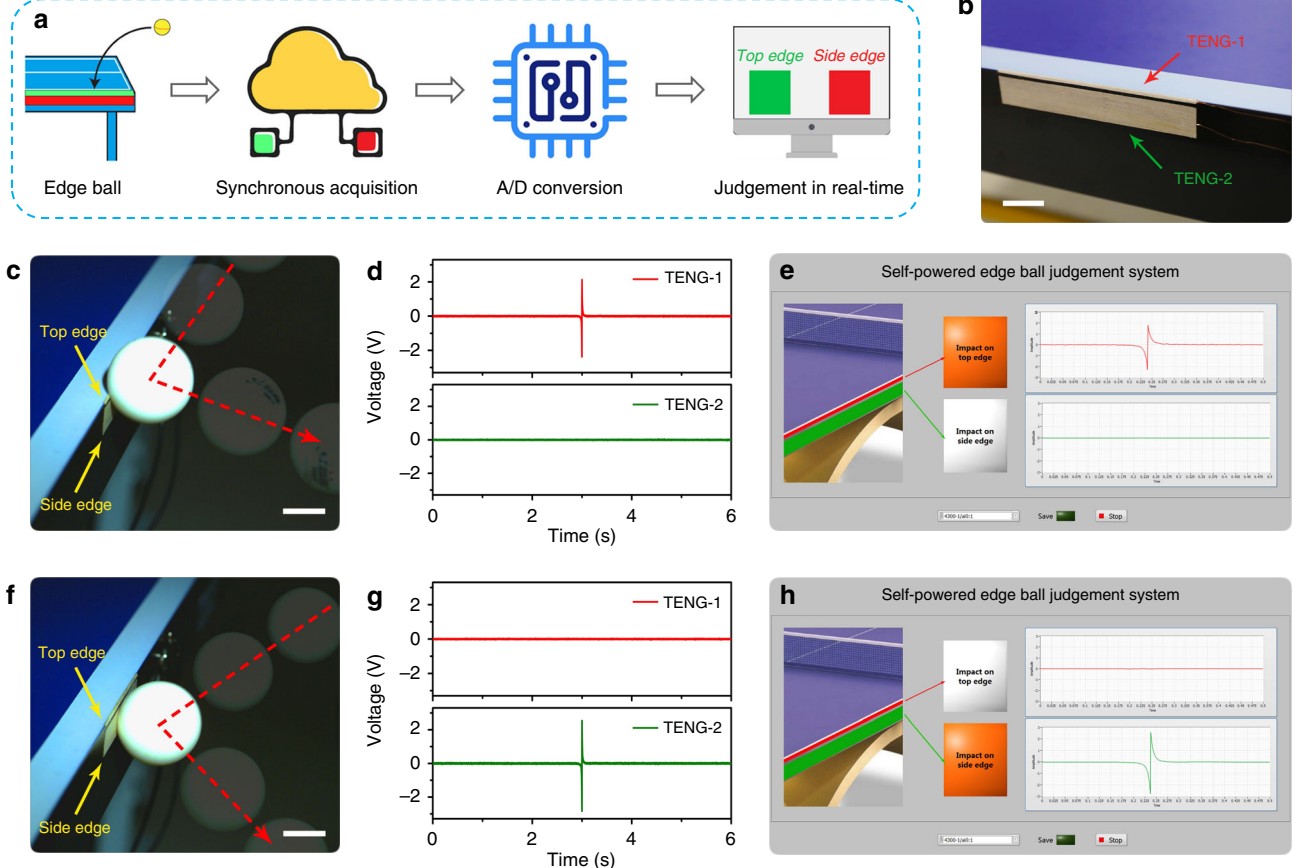

**Fig. 5** Application of the W-TENG in a self-powered edge ball judgement system. **a** Scheme diagram of the W-TENG based self-powered edge ball judgement system. **b** Photograph of two W-TENGs attached on the edge of the ping-pong table. Scale bar, 2 cm. **c–e** Demonstration of the self-powered edge ball judgement system at the moment of top edge ball appeared: **c** Photograph, **d** output signals of two W-TENGs, and **e** screenshot of the real-time judgement result showing a ping-pong ball impacted on the top edge of the table. Scale bar, 2 cm. **f–h** Demonstration of the self-powered edge ball judgement system at the moment of side edge ball appeared: **f** Photograph, **g** output signals of two W-TENGs, and **h** screenshot of the real-time judgement result showing a ping-pong ball impacted on the side edge of the table. Scale bar, 2 cm

wood samples were conducted using a multi-function tribometer (UMT-TriboLab, Bruker) with a steel ball (2.0 mm in diameter) under a vertical force of 2 N.

## Data availability

All data needed to evaluate the conclusions in the paper are present in the paper and/or the Supplementary Information. Additional data related to this paper may be requested from the authors. The source data underlying all figures can be found in the Source Data file.

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

## Acknowledgements

This research was supported by the National Key R&D Project from Minister of Science and Technology (Grant No. 2016YFA0202704), National Natural Science Foundation of China (Grant Nos. 51605033, 51702018, 51432005, and 5151101243), Beijing Municipal Science and Technology Commission (Grant Nos. Z171100002017017, Y3993113DF), China Postdoctoral Science Foundation (Grant No. BX20190324), and the University of Chinese Academy of Sciences. We also thank Prof. Chi Zhang, Ding Li, Chan Wang, and Chao Yuan for helpful discussions and assistance in experiments.

## Author contributions

J.L., F.R.F., and Z.L.W. conceived the idea and designed the experiment. J.L., K.H. and Q.L carried out the wood treatment, materials characterization, and mechanical measurements. J.L. fabricated the devices. J.L., Z.W. and L.X. performed the electrical measurement and Supplementary Movies. T.J. helped with the COMSOL simulations. Y.B. and W.T. provided assistance with the experiments. J.L., L.X., A.C.W., F.R.F. and Z.L.W. wrote the manuscript. All the authors discussed the results and commented on the manuscript.

## Competing interests

The authors declare no competing interests.
