## [Peer Review File · Nature Communications]

Reviewers' comments:

Reviewer #1 (Remarks to the Author):

In the manuscript, the authors reported a flexible, durable and high output wood-based triboelectric nanogenerator (W-TENG) for self-powered sensing application. They modified 'natural wood' to get high-performance and flexible 'treated wood' through simple chemical/physical process. And they demonstrated their enhanced mechanical property as well as high triboelectric property. Thereby, they applied array of W-TENG to self-powered sensor (W-TES, Wood based triboelectric sensor) for sports equipment and athletic big data analytics including self-powered edge ball judgement system. Despite they described the novelty of mechanical and triboelectric property of treated wood and fancy applications, the manuscript has several deficiencies which are recommend to be revised. Therefore, I would recommend this manuscript for the possible publication in Nature Communications after major revision.

The drawbacks and deficiencies are listed below.

Comment 1:

The authors explained that wood has several advantages as an excellent triboelectric material to be applied to sports equipment. According to the description, it is because natural wood is one of the most abundant resources on earth and has been widely used in sports industry. However, almost of wood used in sports equipment is coated with paint or other materials because wood is vulnerable to humidity or getting wet thereby easily permanently deformed. Also, transferred charge density of treated wood is merely $36 \mu\text{C}/\text{m}^2$ and it is too small comparing with other triboelectric materials despite its enhancement of 70 %. Considering features of industry and the result of evaluations, it is regarded that enhancements of mechanical or triboelectric properties of wood are not meaningful. In summary, the authors should describe the fundamental reasons why wood is suitable material to be applied to the triboelectric nanogenerators/ sensors and applications.

Comment 2:

The demonstrations of velocity sensing using W-TES are illustrated in Fig. 4d and supplementary figure 8. However, it is a common knowledge that the output voltage is variable according to the kind of material and environmental factors such as humidity and temperature. Therefore, it is very questionable whether W-TES can exactly detect the velocity of the ball. The authors should prove uniform velocity-sensing ability of the W-TES when the kind of materials and environmental factors are varied.

Comment 3:

Natural wood has very diverse mechanical and electrical property which differs from the kind of wood. I wonder that all of the natural wood can have enhanced properties through two-step process. Also the authors did not describe the kind of wood in detail. Natural wood is merely not a kind of material but a group of materials, the authors should prove the universality of the experimental results. In different ways, the authors should describe the exact kind of wood.

Comment 4:

According to the manuscript, the Young's modulus and coefficient of friction of treated wood apparently differ from those of natural wood. (Fig. 2e) However, the authors described "The wood material in W-TENG can keep the basic properties of the table surface which is significant for sports apparatus." It is an obvious contradiction. As they already mentioned, the properties of surface such as elastic property are very important issues in manufacturing of sports equipment. Despite these changes in surface properties should be minimized from manufacturing point of view, there is no convincing evidences for that.

Comment 5:

Interferometer images are illustrated in supplementary figure 5. However, it would be better to equalize the scale for the exact comparison.

Reviewer #2 (Remarks to the Author):

This manuscript reports a flexible and durable wood-based triboelectric nanogenerator (W-TENG) for self-powered sensing in sports equipment and athletic big data analytics. It is a well-organized manuscript which contains material improvement, device design, and application demonstration. A simple and effective two-step wood treatment method was developed for fabricating wood films with excellent mechanical and triboelectric properties, enabling flexible, durable, and high-performance W-TENGs. Benefiting from such excellent features, the W-TENG is further utilized as a self-powered wood-based triboelectric sensor for fabricating a smart ping-pong table. The authors also demonstrate a series of impressive functional capabilities of the self-powered table tennis training and assisting system, including falling point distribution statistics for athletic big data analytics and disputed edge ball judgement. This research could lead to significant advances in athletic big data analytics and open up a new field of wood-based electronics combining with the self-powered system, and will certainly attract the attention of the broad readership of Nature Communications. The manuscript is well written. Thus, I would like to recommend it to be accepted for publication after proper revision.

1. The smart ping-pong table possesses two capabilities, including distribution statistics and edge ball judgement. However, it is not very clear in Figure 1b. This figure should be modified to make it easier for readers to understand.
2. The weight of the W-TENG is closely related to its size. Although there is a scale bar in Figure 1e, the size of the TENG device had better be add in the manuscript.
3. In Figure 2i and 4e, how many data points were evaluated for the error bars? Please add relevant description in the manuscript.
4. Is the output performance of the W-TENG related to the contact materials? Additional data are needed for this point.
5. For the falling point distribution statistical system, the current size of each sensing unit (4.5×4.5 cm²) is relatively large. How to improve the resolution of the sensing array for large-scale applications?
6. The ball velocity sensing is useful for athletic training. How the ball approaching velocities are measured? The manuscript lacks details about this part, which is essential and should be included in the experimental section.

Manuscript number: NCOMMS-19-05562A

Title: " **Flexible, durable wood based triboelectric nanogenerators for self-powered sensing in sports equipment and athletic big data analytics** "

Authors: Jianjun Luo, Ziming Wang, Liang Xu, Aurelia Chi Wang, Kai Han, Tao Jiang, Qingsong Lai, Yu Bai, Wei Tang, Feng Ru Fan, Zhong Lin Wang

Responses to the reviewers:

Reviewer #1

Comments:

In the manuscript, the authors reported a flexible, durable and high output wood-based triboelectric nanogenerator (W-TENG) for self-powered sensing application. They modified 'natural wood' to get high-performance and flexible 'treated wood' through simple chemical/physical process. And they demonstrated their enhanced mechanical property as well as high triboelectric property. Thereby, they applied array of W-TENG to self-powered sensor (W-TES, Wood based triboelectric sensor) for sports equipment and athletic big data analytics including self-powered edge ball judgement system. Despite they described the novelty of mechanical and triboelectric property of treated wood and fancy applications, the manuscript has several deficiencies which are recommend to be revised. Therefore, I would recommend this manuscript for the possible publication in Nature Communications after major revision.

The drawbacks and deficiencies are listed below.

Response: Thank you very much for your positive comments on the manuscript. We have revised the manuscript carefully according to your suggestions, and all the revisions are highlighted with yellow background in the revised manuscript.

1. The authors explained that wood has several advantages as an excellent triboelectric material to be applied to sports equipment. According to the description, it is because natural wood is one of the most abundant resources on earth and has been widely used in sports industry. However, almost of wood used in sports equipment is coated with paint or other materials because wood is vulnerable to humidity or getting wet thereby easily permanently deformed. Also, transferred charge density of treated wood is merely $36 \mu\text{C}/\text{m}^2$ and it is too small comparing with other triboelectric materials despite its enhancement of 70 %. Considering features of industry and the

result of evaluations, it is regarded that enhancements of mechanical or triboelectric properties of wood are not meaningful. In summary, the authors should describe the fundamental reasons why wood is suitable material to be applied to the triboelectric nanogenerators/ sensors and applications.

Response: Thank you for the suggestion. Currently, most of the materials used for TENGs are non-degradable synthetic polymers, which could possibly result in severe environmental pollutions. Natural wood is one of the most abundant resources on earth. As a renewable, sustainable, and biodegradable material, it has been widely used in buildings, energy storage, and water purification. Via advanced wood treatment strategies, different novel wood-based materials have been designed, such as cooling wood¹, transparent wood², conductive wood^{3,4}. However, the mechanical and triboelectric performance of natural wood is unsatisfactory for fabricating TENGs. Thus, an effective processing method that can adjust the properties of the natural wood for fabricating high-performance TENG will provide exciting opportunities for developing sustainable and eco-friendly self-powered systems. In this work, we developed a new flexible, durable wood-based TENG and apply it in constructing intelligent sports equipment for the first time. It is hopeful to open up a new field of wood-based electronics combining with the self-powered system.

After chemical treatment and hot pressing, the moisture resistance of the treated wood can be greatly improved, as shown in Supplementary Figure 6. After sustaining 95% relative humidity (RH) for 12 hours, the thickness swelling rate of the pressed natural wood reaches 366.7%. However, the thickness of the treated wood remains unchanged even after sustaining 95% RH for 240 hours, demonstrating its excellent ability in anti-moisture. For future development, the moisture resistance of the treated wood can be further improved through surface treatment methods.

For self-powered sensing, the output performance of the TENG doesn't have to be very high. Owing to the output enhancement of the treated wood, the sensitivity of the W-TES can be greatly improved.

To sum up, the wood is a suitable material to be applied for constructing TENGs and self-powered sensors. We wish the above could respond to your concerns. Thank you very much.

Supplementary Figure 6. Dimensional stability of the pressed natural wood and treated wood against moisture. (a) Change in thickness of the pressed natural wood and treated wood under 95% RH over time. (b) Percentage increase in thickness of the pressed natural wood and treated wood after sustaining 95% RH for 12 hours. (c) Long-term change in thickness of the treated wood under 95% RH for 240 hours.

2. The demonstrations of velocity sensing using W-TES are illustrated in Fig. 4d and supplementary figure 8. However, it is a common knowledge that the output voltage is variable according to the kind of material and environmental factors such as humidity and temperature. Therefore, it is very questionable whether W-TES can exactly detect the velocity of the ball. The authors should prove uniform velocity-sensing ability of the W-TES when the kind of materials and environmental factors are varied.

Response: Thank you for the suggestion. While the surfaces of two different materials are brought into physical contact, triboelectric charges will be created on the two contacted surfaces. Based on this working mechanism, the TENG-based self-powered sensor will be applicable to various kinds of materials. As shown in Supplementary Figure 13, the output voltage of the W-TENG when contacting with different materials are tested, which depends on the relative ability of a dielectric

material to gain electrons when contacting with the treated wood. Since the sensing mechanism of the W-TES is consistent regardless of the change in contact materials, its output signal will still increase along with the increase of the approaching velocity between the moving object and the W-TES. The type of material can be a calibration parameter for the velocity sensing.

Besides, the influences of environmental humidity on the W-TENG are also investigated. First, as mentioned above, the structure of treated wood would not be influenced by the environmental humidity (Supplementary Figure 6). Second, the output voltage of the W-TENG in different relative humidity has also been studied, as shown in Supplementary Figure 12. It could be seen that the output voltage of the W-TENG will decrease with the increase of the RH. However, this overall output decline will not influence the velocity-sensing ability of the W-TES. Through appropriate calibration of output decline caused by RH increasing, the W-TES can still work normally as a self-powered velocity sensor.

The temperature-dependence of contact-electrification has been studied in our previous research^{5,6}. It has been proved that the output performance of the TENG is relatively stable at general ambient temperature (0-40 °C, definitely including the temperature range for sports competition). Although the output signal of the W-TES will decline slightly as the temperature rises, it can also be used for ball velocity sensing at ambient temperature through appropriate calibration.

In summary, the W-TES could be effectively used for velocity-sensing when the kind of materials and environmental factors are varied.

Supplementary Figure 13. Summarized open-circuit voltage of a W-TENG with relative contact-separation motion to different materials. Error bars indicate standard deviations for 3 sets of data points.

Supplementary Figure 12. Dependence of the open-circuit voltage of the W-TENG on RH.

3. Natural wood has very diverse mechanical and electrical property which differs from the kind of wood. I wonder that all of the natural wood can have enhanced properties through two-step process. Also the authors did not describe the kind of wood in detail. Natural wood is merely not a kind of material but a group of materials, the authors should prove the universality of the experimental results. In different ways, the authors should describe the exact kind of wood.

Response: Thank you very much for your suggestion. In this work, balsa wood was chosen for two-step processing and constructing the W-TENG. We are sorry that this part of information is not detailed enough. Relevant description has been added in the revised manuscript. The main components of various kinds of natural woods are cellulose, hemicellulose and lignin. Our two-step wood treatment method includes a boiling process in aqueous mixture of NaOH and Na₂SO₃ followed by hot-pressing. Chemical treatment leads to substantial reduction of lignin/hemicellulose content, but only modest reduction of cellulose. By partial removal of lignin/hemicellulose from the wood cell walls, the wood becomes more porous and less rigid, which is beneficial and essential to the following hot-pressing process for further TENGs fabrication. Thus, this processing method should be universally effective for various species of wood and can greatly enhance their mechanical and triboelectric properties simultaneously.

To prove the universality of our two-step wood treatment process, different kinds of the natural wood (basswood, pine, oak) were used as triboelectric materials for

constructing W-TENG. As shown in Supplementary Figure 9, the transferred charge density of the W-TENGs based on various kinds of treated woods are all much higher than that of the natural woods, demonstrating the universality and effectiveness of our wood treatment strategy. This could be due to the similar chemical composition change of various kinds of natural woods. As regard to the basic mechanical properties improvement, previous research has proved the universality of the two-step wood treatment method⁷. Relevant descriptions have been added in the revised version of the manuscript.

Supplementary Figure 9. Comparison of the transferred charge density between W-TENGs based on various species of natural wood and treated wood. Error bars indicate standard deviations for 3 sets of data points.

4. According to the manuscript, the Young's modulus and coefficient of friction of treated wood apparently differ from those of natural wood. (Fig. 2e) However, the authors described "The wood material in W-TENG can keep the basic properties of the table surface which is significant for sports apparatus." It is an obvious contradiction. As they already mentioned, the properties of surface such as elastic property are very important issues in manufacturing of sports equipment. Despite these changes in surface properties should be minimized from manufacturing point of view, there is no convincing evidences for that.

Response: Thank you for the comment. In the sentence of "The wood material in W-TENG can keep the basic properties of the table surface, which is significant for sports apparatus", we would like to indicate that the W-TENG installing on the ping-pong table will not change the surface material composition, which is significant for building smart sports apparatus. We are sorry for the confusion. This sentence has

been changed to “The wood material in W-TENG can keep the basic material composition of the table surface, which is significant for building smart sports apparatus” in the revised version of the manuscript.

Actually, owing to the mechanical properties enhancement of the treated wood, the surface properties of the smart sports equipment can be further improved. First, we have investigate the influence of the W-TENG on the elastic property of the table surface, as shown in Supplementary Figure 14. When a standard ping-pong ball is dropped on to the surface of the ping-pong table from a height of 30 cm, a rebound height of 24.6 cm can be measured. While dropping on the W-TENG based on the treated wood attaching on the table surface, the rebound height only slightly decrease to about 24.0 cm, which meets the laws of table tennis set by the International Table Tennis Federation (ITTF). The maintaining of surface elastic property could be explained by the great hardness enhancement of the treated wood. However, the rebound height reduces to 20.7 cm once dropping on the W-TENG based on the natural wood, which is much lower than the standard set by the ITTF. Relevant data have been added in the revised version of the manuscript. Besides, compared with the natural wood, the tensile strength and wear resistance enhancement of the treated wood can improve the stability and durability of the sports equipment. Moreover, benefit from its excellent flexibility, the treated wood can easily fit on the edge of the table.

Supplementary Figure 14. Influence on the elastic property of the table surface by the W-TENG based on natural wood and treated wood. The rebound height is measured by dropping a standard ping-pong ball onto the surface of the table. Error bars indicate standard deviations for 3 sets of data points.

5. Interferometer images are illustrated in supplementary figure 5. However, it would be better to equalize the scale for the exact comparison.

Response: Thank you very much for your reminder. We have equalized the scales in each of the figures in Supplementary Figure 5.

Supplementary Figure 5. Interferometer images showing the surface topography of the treated wood after hot pressing using various sand papers as the mold. (a) control. (b) 2000-grit. (c) 1000-grit.

Reviewer #2

Comments:

This manuscript reports a flexible and durable wood-based triboelectric nanogenerator (W-TENG) for self-powered sensing in sports equipment and athletic big data analytics. It is an well-organized manuscript which contains material improvement, device design, and application demonstration. A simple and effective two-step wood treatment method was developed for fabricating wood films with excellent mechanical and triboelectric properties, enabling flexible, durable, and high-performance W-TENGs. Benefiting from such excellent features, the W-TENG is further utilized as a self-powered wood-based triboelectric sensor for fabricating a smart ping-pong table. The authors also demonstrate a series of impressive functional capabilities of the self-powered table tennis training and assisting system, including falling point distribution statistics for athletic big data analytics and disputed edge ball judgement. This research could lead to significant advances in athletic big data analytics and open up a new field of wood-based electronics combining with the self-powered system, and will certainly attract the attention of the broad readership of Nature Communications. The manuscript is well written. Thus, I would like to recommend it to be accepted for publication after proper revision.

Response: Thank you very much for your positive comments on the manuscript. We have revised the manuscript carefully according to your suggestions, and all the revisions are highlighted with yellow background in the revised manuscript.

1. The smart ping-pong table possesses two capabilities, including distribution statistics and edge ball judgement. However, it is not very clear in Figure 1b. This figure should be modified to make it easier for readers to understand.

Response: Thank you for the suggestion. To emphasize the two capabilities of the smart ping-pong table, we have added relevant arrows and sequence numbers in Figure 1b. We hope this can make it easier to read and comprehend.

Fig. 1 Fabrication process and schematic illustration of the flexible wood-based TENG (W-TENG) and smart ping-pong table. **a** Diagram of the process for fabricating a flexible W-TENG. **b** Schematic illustration of the natural wood-based smart ping-pong table. **c** Scanning electron microscopy (SEM) image of the treated wood surface. Scale bar, 20 μm . **d** Photograph of the as-prepared W-TENG devices, demonstrating its excellent flexibility. Scale bar, 2 cm. **e** Weight of the W-TENG device. Scale bar, 3 cm.

2. The weight of the W-TENG is closely related to its size. Although there is a scale bar in Figure 1e, the size of the TENG device had better be add in the manuscript.

Response: Thank you very much for the reminder. With a weight of 190 mg, the size of the W-TENG device in Figure 1e is $4 \times 4 \text{ cm}^2$. We have added relevant description in the revised manuscript.

3. In Figure 2i and 4e, how many data points were evaluated for the error bars? Please add relevant description in the manuscript.

Response: Thank you for the suggestion. In Figure 2i and 4e, the error bars were evaluated by 5 sets of data points. Relevant descriptions have been added in the revised manuscript.

4. Is the output performance of the W-TENG related to the contact materials? Additional data are needed for this point.

Response: Thank you for the suggestion. Because of the universality of contact electrification between any two different materials, the W-TENG can generate outputs from relative motion between the W-TENG and many other materials. A series of materials were used to have contact-separation motion relative to the W-TENG in single-electrode mode, and corresponding open-circuit voltages were recorded, as shown in Supplementary Figure 13. The output depends on the relative ability of a dielectric material to gain electrons when contacting with the treated wood, coincident with the well-established tribo-series table^{8,9}. Relevant descriptions have been added in the revised manuscript.

Supplementary Figure 13. Summarized open-circuit voltage of a W-TENG with relative contact-separation motion to different materials. Error bars indicate standard deviations for 3 sets of data points.

5. For the falling point distribution statistical system, the current size of each sensing unit ($4.5 \times 4.5 \text{ cm}^2$) is relatively large. How to improve the resolution of the sensing array for large-scale applications?

Response: Thank you for the suggestion. There are two methods to improve the resolution of the sensing array. The first one is to reduce the size of the W-TES unit, which can be easily cut into smaller size. The second one is to optimize the wiring design of the W-TES array. The metal wires attaching to the electrode of the W-TES could be set under the table from manufacturing point of view, therefore the gap between two adjacent W-TESs can be reduced. We have added relevant descriptions in the revised manuscript.

6. The ball velocity sensing is useful for athletic training. How the ball approaching velocities are measured? The manuscript lacks details about this part, which is essential and should be included in the experimental section.

Response: Thank you very much for your precious reminder. The ping-pong ball was dropped from various heights to achieve different ball approaching velocities through free-falling method. And these velocities were calculated via the following expression without taking the effect of air damping into consideration:

$$v = \sqrt{2gh} \quad (1)$$

In this equation, v represents the approaching velocity, g is the local gravitational acceleration, and h is the falling height. Relevant descriptions have been added in Supplementary Note 2. Besides, the approaching velocities of the ping-pong ball were also verified by high speed camera measurement.

References

- 1 Li, T. *et al.* A radiative cooling structural material. *Science* **364**, 760-763 (2019).
- 2 Zhu, M. *et al.* Highly anisotropic, highly transparent wood composites. *Adv. Mater.* **28**, 5181-5187 (2016).
- 3 Chen, C. *et al.* All-wood, low tortuosity, aqueous, biodegradable supercapacitors with ultra-high capacitance. *Energy Environ. Sci.* **10**, 538-545 (2017).
- 4 Ye, R. *et al.* Laser-induced graphene formation on wood. *Adv. Mater.* **29**, 1702211 (2017).
- 5 Xu, C. *et al.* On the Electron-Transfer Mechanism in the Contact Electrification Effect. *Adv. Mater.* **30**, 1706790 (2018).
- 6 Wen, X., Su, Y., Yang, Y., Zhang, H. & Wang, Z. L. Laser-induced graphene formation on wood. *Nano Energy* **4**, 150-156 (2014).
- 7 Song, J. *et al.* Processing bulk natural wood into a high-performance structural material. *Nature* **554**, 224-228 (2018).
- 8 Davies, D. K. Charge generation on dielectric surfaces. *J. Phys. D Appl. Phys.* **2**, 1533-1537 (1969).
- 9 Diaz, A. F., & Felix-Navarro, R. M. A semi-quantitative tribo-electric series for polymeric materials: the influence of chemical structure and properties. *J. Electrostat.* **62**, 277-290 (2004).

REVIEWERS' COMMENTS:

Reviewer #1 (Remarks to the Author):

The authors took into consideration the reviewer's comments, they analyzed additional measurements and answered to the questions. They addressed most of reviewers' comments and took them into account by modifying the manuscript and the supporting information. They nicely answered the questions by providing data, analysis and discussions. The manuscript has been changed and is now better and clearer. Overall I think this work can now be published in Nature Communications. This is why I recommend accepting this manuscript.

Reviewer #2 (Remarks to the Author):

This revised paper can satisfactorily answer these comments. Therefore, it can be accepted as it is!

Manuscript number: NCOMMS-19-05562A

Title: " **Flexible, durable wood based triboelectric nanogenerators for self-powered sensing in sports equipment and athletic big data analytics** "

Authors: Jianjun Luo, Ziming Wang, Liang Xu, Aurelia Chi Wang, Kai Han, Tao Jiang, Qingsong Lai, Yu Bai, Wei Tang, Feng Ru Fan, Zhong Lin Wang

Responses to the reviewers:

Reviewer #1

Comments:

The authors took into consideration the reviewer' s comments, they analyzed additional measurements and answered to the questions. They addressed most of reviewers' comments and took them into account by modifying the manuscript and the supporting information. They nicely answered the questions by providing data, analysis and discussions. The manuscript has been changed and is now better and clearer. Overall I think this work can now be published in Nature Communications. This is why I recommend accepting this manuscript.

Response: Thank you very much for your positive comments on the manuscript.

Reviewer #2

Comments:

This revised paper can satisfactorily answer these comments. Therefore, it can be accepted as it is!

Response: Thank you very much for your positive comments on the manuscript.